# Emotion Recognition in a Health Continuum: Comparison of Healthy Adults of Advancing Age, Community Dwelling Adults Bearing Vascular Risk Factors and People Diagnosed with Mild Cognitive Impairment

**DOI:** 10.3390/ijerph192013366

**Published:** 2022-10-16

**Authors:** Glykeria Tsentidou, Despina Moraitou, Magdalini Tsolaki

**Affiliations:** 1Laboratory of Psychology, Department of Experimental and Cognitive Psychology, School of Psychology, Aristotle University of Thessaloniki, 54124 Thessaloniki, Greece; 2Laboratory of Neurodegenerative Diseases, Center for Interdisciplinary Research and Innovation (CIRI), Aristotle University of Thessaloniki, 54124 Thessaloniki, Greece; 3Greek Association of Alzheimer’s Disease and Related Disorders (GAADRD), 54643 Thessaloniki, Greece

**Keywords:** basic emotions, cognitive impairment, dynamic visual displays, educational level, vascular burden

## Abstract

The identification of basic emotions plays an important role in social relationships and behaviors linked to survival. In neurodegenerative conditions such as Alzheimer’s disease (AD), the ability to recognize emotions may already be impaired at early stages of the disease, such as the stage of Mild Cognitive Impairment (MCI). However, as regards vascular pathologies related to cognitive impairment, very little is known about emotion recognition in people bearing vascular risk factors (VRF). Therefore, the aim of the present study was to examine emotion recognition ability in the health continuum “healthy advancing age—advancing age with VRF—MCI”. The sample consisted of 106 adults divided in three diagnostic groups; 43 adults with MCI, 41 adults bearing one or more VRF, and 22 healthy controls of advancing age (HC). Since HC were more educated and younger than the other two groups, the age-group and level of educational were taken into account in the statistical analyses. A dynamic visual test was administered to examine recognition of basic emotions and emotionally neutral conditions. The results showed only a significant diagnostic group x educational level interaction as regards total emotion recognition ability, F (4, 28.910) = 4.117 *p* = 0.004 η^2^ = 0.166. High educational level seems to contribute to a high-level-emotion-recognition-performance both in healthy adults of advancing age and in adults bearing vascular risk factors. Medium educational level appears to play the same role only in healthy adults. Neither educational level can help MCI people to enhance their significantly lower emotion recognition ability.

## 1. Introduction

Emotion recognition -ER- is defined as the ability to classify and recognize an emotional state from sensory cues [1], which is closely associated with the quality of daily life and survival [2,3]. Recognition of others’ emotions is considered a main component of social cognition [4] and the deficit in this ability causes adverse effects in communication, low social competency and often leads to isolation and depression [5]. 

The assessment of emotion recognition can be carried out in two ways: using static images or dynamic stimuli—morphed photographs and/or videos [6]. The differences between the two methods do not solely pertain to the types of stimuli they use, but also to their cognitive demands [7]. It, therefore, appears that emotion recognition performance with dynamic stimuli is more related to mental speed, attention, and working memory [8] and the neural areas they mediate. [9]. It has also been shown that even the same emotions triggered by different stimuli (static or dynamic) may involve different brain regions. A prominent example is that of anger, which, when triggered by a dynamic stimulus, is mediated by increased right-lateralized activity in the medial, superior, middle, and inferior frontal cortex and cerebellum. However, if triggered by a static stimulus, it activates a motor, prefrontal, and parietal cortical network [10].

McDonalds et al. [11] examined the different demands of dynamic stimuli using the tool they designed, The Awareness of Social Inference Test; dynamic audiovisual displays, conversational tone alone, moving facial displays, and still photographs in different ways. Their findings show the mediation of different and individual brain structures for the said different demands. Apart from these differences in brain channels [12] used for the dynamic stimuli demands, they also each involve different cognitive abilities [13]. 

To conclude, dynamic stimuli appear to be mediated by brain regions related to social aspects and emotional processing [14]. Furthermore, as previously stated, they require greater cognitive reserves, whereas it has been supported that due to their said characteristics, they have greater ecological validity, as compared to static stimuli, regarding the assessment of emotion recognition ability [15,16].

Besides potential stimuli differences, it is common acceptance that the neurobiological substrates of ER include the amygdala and superior temporal sulcus/gyrus [17], as well as fusiform gyri, somatosensory, medial prefrontal and cingulate areas [18]. More recently, Aben (2020) [1] argues that in the debate between which hemisphere prevails in emotion recognition, the involvement of both hemispheres is confirmed [19,20]. The same author [1] confirms the neuronal involvement of the medial prefrontal cortex, the anterior cingulate cortex and the insula in the network of ER. 

It is worth noting that in recent years there have been added to the literature studies on the recognition of emotions by investigating them through electroencephalograms, recording differences in the ability to recognize emotions as well as differences in mediating areas. [21,22]. The results of studies of the specific methods find space in important fields such as emotional calculation, artificial intelligence, learning and rehabilitation [23,24,25].

Reviewing the areas which mediate ER, it is understood that ER will probably be disturbed in persons with several neurological and psychiatric disorders. More specifically, for neurodegenerative disorders—Alzheimer’s disease [26,27] Parkinson [28,29] and Frontotemporal dementia [30], ER seems to function as an important criterion for the diagnosis and progression of the disease. Regarding AD, the findings agree that ER deficits play a particular role in the progression of dementia [5,31,32]. Key conclusions include impaired ability to recognize negative emotions, especially at the onset of the disease [33], while in the course of the disease the weaknesses seem to generalize to other emotions [26,34]. Lesions of areas which are associated with the disease and mediate the recognition of emotions—amygdalar volume, volume of the pallidum and of the fusiform gyrus [35] are included in the interpretation of the said deficits. In the behavioral level, ER deficits are correlated with alterations in attention and processing speed, in long-term memory, visuospatial dysfunctions or social and behavioral difficulties faced by patients with Alzheimer’s dementia [2]. 

On the other hand, data for MCI—predementia stage with objective cognitive and memory deficits—although not severe enough to meet the dementia criteria [36] are presented divided. Elferink et al. (2015) [27] emphasize deficits in recognizing negative emotions in patients with MCI, while McCade et al. [37] argue that deficits are observed only in amnestic subtype (aMCI). Specifically, a recent study [28] showed that defective performance in ER tasks, may be a useful cognitive marker to detect limbic-predominant aMCI people among the heterogeneous aMCI population. Earlier, Phillips et al. 2003 [38] supported that ER relies on the ventral affective system, including the amygdala, insula, ventral striatum, and ventral regions of the anterior cingulate gyrus and prefrontal cortex, areas that are generally early affected in MCI [39]. In any case, Mc Cade [37] underlined that research on ER in this field is limited and the variability in findings is large, emphasizing the need for further research.

In a health continuum from ‘absolute health’ to ‘death’, before MCI, a ‘stage’ of experiencing vascular aging and subtle cognitive impairment could be proposed, and, in this vein, recent research has shown that non-diagnosed vascular pathology and vascular pathology related to vascular risk factors is indeed linked to cognitive impairment [40,41,42]. In other words, it has already been argued that subclinical diseases, such as hypertension or diabetes that are common among community dwelling older adults, are linked to impaired cognition. Indeed, recent studies have shown that cerebral small vessel disease (SVD) is recognized as a major contributor in the pathogenesis of neurodegeneration, including AD, and is closely associated with modifiable vascular risk factors, such as hypertension. Hence, SVD affecting perforating vessels in the brain, may function as a main dynamic underlying brain pathology that links the stage of experiencing vascular risk factors to the next stages of developing AD pathology [42,43,44,45].

The association of vascular risk factors and the resulting brain alterations with the ability to recognize emotions, are found either in vascular type dementias [46] or in ischemic brain injuries [47] or in aging [3], which is usually accompanied by the aforementioned vascular risk factors (vascular aging). An attempt to group the findings demonstrates that the deficits of adults with cerebrovascular accidents and vascular type dementia disorders in emotion recognition are related to the general challenges in global cognition [48,49,50]. Concerning vascular aging, it is mainly explained by brain lesions linked to aging—lesions in frontal and temporal regions [51].

### The Purpose and the Hypotheses of the Study

In this context, the aim of the present study was to examine the ability of emotion recognition in the health continuum “healthy advancing age—advancing age with VRF—MCI”, comparing the performance of people in these three diagnostic groups in emotion recognition (ER) using dynamic visual displays.

Based on the literature, the hypotheses of the study were formulated as follows: (a) MCI patients would show lower performance in ER—due to possible brain pathology of AD type and especially of aMCI type, or to the combination of this pathology and vascular pathologies, compared to the other two groups; (b) community dwelling adults bearing VRF would perform at a lower level—due to a potential underlying brain pathology which is accelerated by the risk factors (e.g., small vessels disease), compared to Healthy Controls.

## 2. Methods

### 2.1. Participants

Τhe sample was convenient. We recruited a total of 106 adults from the “Alzheimer Hellas” Memory Clinics in Thessaloniki, Greece, from outpatient clinics of the General Hospital of Katerini, Greece, and a sample of healthy adult volunteers from the wider community of Pieria. Their age ranged from 50 to 85 years (M = 66.09, SD = 9.02). The sample did not include adults with mood and/or anxiety disorders, neurological disorders of any type, dementia of any type, patients diagnosed with cancer within the last five years, adults after stroke, myocardial infarction and cardiac instabilities, or patients having undergone any other surgery within the last five years. Adults with mental and/or psychiatric diseases and adults with alcoholism and/or drug use disorders were excluded from the study sample.

All participants were screened for depressive symptomatology using the Geriatric Depression Scale-15 [52,53], and persons with scores > 6 were excluded from the study. To assess the ability of simple and complex sentence comprehension, we used the subscale “auditory perception” from the Boston Diagnosing Aphasia Examination [54]. To assess general cognitive ability, the Montreal Cognitive Assessment (MoCA) Scale was selected [55,56,57]. Moreover, standardized tests were administered for the assessment of general cognitive and functional abilities, memory capacity, attention, language abilities, and executive functions. The entirety of the neuropsychological tests included in this battery is presented in detail in Tsolaki et al. (2017) [57].

MCI group. In detail, the first group consisted of 43 adults (9 men and 34 women, Mean Age = 70.2, SD = 7) diagnosed with MCI during the last two years. Τhe inclusion criteria were based on the DSM-5 criteria for Mild Neurocognitive Disorders (American Psychiatric Association, 2013). Their diagnosis, besides the neuropsychological assessment, was supported by neurological examination, neuropsychiatric assessment, and blood tests. The inclusion criteria included: (a) diagnosis of Minor Neurocognitive Disorders, according to DSM-5, (b) 1.5 standard deviation (SD) below the normal mean, in at least one cognitive domain, according to the neuropsychological tests. According to the evaluations, the vast majority (over 80%) of MCI participants were of the amnestic type with multiple domains. Τheir MoCA test ranged from 21 to 29 (M = 24.4, S.D. = 2.1). At this point, it is underlined that 35 MCI patients had at least one of the vascular risk factors examined in this study (hypertension, hyperlipidemia, diabetes mellitus) and were receiving the appropriate medication. Regarding their educational level, 15 participants had a low level of education (0–9 years), 11 had a medium level (10–12 years), and 17 were highly educated (13 years and over) based on the number of years of education. As for their age-group, 12 MCI participants were late middle-aged adults (50–64 years of age) and 31 were older adults (65–85 years of age) (Table 1).

VRF group. The second group consisted of older adults who visited the outpatient clinics at the General Hospital of Katerini for drug prescription only (related to VRF), without any diagnosis related to cognitive decline. They were under medical supervision and medication due to vascular risk factors (n = 41, 9 men and 32 women, Mean Age = 68.6, SD = 7). Their blood test taken the previous year reported at least one of three common VRF (hypertension, hyperlipidemia, diabetes mellitus). Exclusion criteria comprised diagnosis of MCI or dementia of any type, and all criteria described above, which were observed for all groups. Their general cognitive status, as measured by the MoCA, ranged from 25 to 30 (M = 26.7, S.D. = 1.4). Their educational level varied; 21 participants had a low educational level, 8 had a medium one and 12 were highly educated. As for the age-group, they were divided; 14 VRF participants were late middle-aged adults (50–64 years of age) and 27 were older adults (65–85 years of age) (Table 1).

Healthy Controls. The control group consisted of healthy community-dwelling adult volunteers with excellent physical and mental health, who were asked not to receive any medication and whose blood check-up in the last six months did not show any VRF. Moreover, it is noted that HC did not report memory complaints (this was examined using one question regarding whether the participants felt their memory worsen compared to five years ago, to which they could reply “yes” or “no”). Τo meet this requirement, the third group consisted of younger adults (M = 54.25, SD = 3.0) as compared to the other two groups; all participants of HC (N = 22) were late middle-aged adults (50–64 years of age). From HC, only 3 participants had a low educational level, 9 had medium educational level, and 10 were highly educated (Table 1). Their MoCA score ranged from 26 to 30 (M = 27.7, SD = 1.3).

Based on the admission criteria for the groups, as presented above, and aiming to instigate the assumption whether the three groups differ from each other in terms of their demographic characteristics, ANOVAs were performed, as regards the exact age and years of education. Hence, the three groups differ significantly in age, F (2, 108) = 52,403 *p* < 0.001, and in years of education, F (2, 108) = 4,150, *p* = 0.018. No difference arose in the gender of the participants, χ^2^ (2, 1) = 0.187 *p* = 0.911, although it should be noted that the female gender was overrepresented in all groups. Reviewing the differences in the groups, it turned out that HC were younger compared to both MCI participants, I-J = −15.955 *p* < 0.001, and VRF adults, I-J = -14.360, *p* < 0.001, respectively. On the other hand, the HC group tended to a slight differentiation in the years of education, as compared only to adults with VRF, I-J = 2.829, *p* = 0.019. MCI and VRF groups did not differ in age and education. 

### 2.2. Procedure

The participants from Katerini (MCI, VRF and HC) were examined at the Day Care Center for Dementia Disease Patients at the General Hospital of Katerini, while participants from Thessaloniki were examined individually at the two “Alzheimer Hellas” Centers. Τhe same author completed the process of allocating every participant in each group, based on medical examinations and diagnoses, medication, comprehensive medical history, and neuropsychological tests, as mentioned before. The main assessments of the study, part of which is the examination of emotion recognition ability, were completed in successive sessions; in the first meeting, participants were evaluated using the selected tools for screening, as described above. Moreover, a comprehensive medical history of their health status was obtained. In the next sessions, participants were extensively assessed for their executive functions, memory capacity, Theory of Mind abilities and emotion recognition ability. It should be noted that in one of these sessions, participants were administered only the test examining emotion recognition using dynamic visual displays.

Ethics. For the purposes of this study, the participants provided written informed consent at the time of their first visit, agreeing to their volunteer participation and their withdrawal at any time, without providing a reason and without cost. The present study is part of the cross-sectional study of the first author’s (Glykeria Tsentidou) doctoral dissertation. The protocol of the study was approved by the Scientific and Bioethics Committee of the Greek Association of Alzheimer’s Disease and Related Disorders (Scientific Committee Approved Meeting Number: 25/21-06-2016) and followed the principles outlined in the Helsinki Declaration of 1975, as revised in 2008. Moreover, the study was approved by the Hellenic Data Protection Authority, License number: 1971.

## 3. Measure

Emotion Evaluation Task (EET) of The Awareness of Social Inference Test (TASIT): The Emotion Evaluation Task (EET) is an audiovisual tool designed for the clinical assessment of basic emotion recognition [58]. The basic idea behind the test is the acceptance that there is a limited number of basic emotions that are common across all cultures and ages—Happiness, Sadness, Anger, Fear-Anxiety, Disgust and Surprise (Ekman 1973 [59]). However, in this test there are no static pictures for these emotions. The emotions are presented dynamically and are associated as closely as possible to everyday life situations. Specifically, the test comprises a video of short scenes of actors interacting in everyday situations. The target actor in each scene exhibits one of the six basic emotions or, in some scenes, the actor does not exhibit any emotion (emotionally neutral condition). After viewing each scene, the participant is instructed to choose the label of the emotion they believe the actor was exhibiting from a multiple-choice array. There are a total of 28 scenes, comprising four examples of each of the six emotional and one non-emotional states [60]. It is noted that in the present study, the video was playing with the sound turned off, taking into account the instructions of the authors who developed the test to scientists who aim to use it in a different language than English. Therefore, we focused on emotion recognition from dynamic visual cues without additional assistance of voice, prosody, or other linguistic elements. 

As regards the psychometric properties of the EET, Moraitou et al. [61] confirmed a uni-factorial structure of the test for the Greek population. The factor labeled “decoding of uncertain emotional displays” was loaded by surprise, sadness, anger, anxiety, and disgust items, while happiness and neutral condition variables were allowed in the Structural Equation Model (SEM) to freely co-vary with the factor, Satorra-Bentler scaled χ^2^ (14, n = 208) = 18.36, *p* = 0.191, CFI = 0.99 and RMSEA = 0.04 (90% CI: 0.00–0.08).

Based on the confirmed SEM model, in the present study we created a variable labeled “total emotion recognition” by summing up the answers of the participants for the surprise, sadness, anger, anxiety, and disgust variables. Besides the general ER variable, we kept the variable “Happiness” and the variable “Neutral condition” separate. 

## 4. Statistical Analysis

The analyses carried out were (a) MANOVA and (b) ANOVA. For the analyses we used the Statistical Package for the Social Sciences [62]. Box’s M Test was used for the assessment of the equivalence of covariance matrices (for the MANOVA). Levene’s test was used to assess the equality of variances (for the ANOVAs). Partial eta-squared (ηp^2^) was used for the estimation of the effect size. Bonferroni correction was adopted for mean comparisons and multiple testing. Given that there were 7 states measured (6 emotional and 1 non-emotional), p was defined as follows: *p* = 0.05/7 = 0.007. 

## 5. Results

Given the composition and the differences of the diagnostic groups as regards individual-demographic factors, a 3 (Diagnostic group: MCI, VRF, HC) × 2 (gender: men, women) × 2 (age-group: late middle-aged adults, older adults) × 3 (educational level: low, medium, high) between-subjects multivariate analysis of variance was performed on the two dependent variables of Happiness and Neutral condition performance. No significant result was found at the multivariate level, F (4.000, 190.000) = 2.270, *p* = 0.063, η^2^ = 0.046.

However, the observation of the ANOVAs showed that in terms of the recognition of the neutral condition, there was a tendency, F (2, 5.330) = 4.376, *p* = 0.015, η^2^ = 0.084, for differentiation of the diagnostic groups. Post hoc Bonferroni comparisons showed that HC tended to significantly differ in their performance with MCI, I-J = 0.76, *p* = 0.035, and VRF, I-J = 1.33, *p* < 0.001. MCI and VRF groups did not differ between each other F (1, 5.335) = 4.380, *p* = 0.039, η^2^ = 0.044.

Figure 1 shows the mean scores of the three diagnostic groups in the two variables. As shown, all diagnostic groups are able to recognize happiness at an almost excellent rate. On the other hand, only HC appear able to detect the non-emotional condition, while MCI and VRF, who are also older than HC, seem to have difficulties. 

Subsequently, a 3 (Diagnostic group: MCI, VRF, HC) × 2 (gender: men, women) × 2 (age-group: late middle-aged adults, older adults) × 3 (educational level: low, medium, high) between-subjects analysis of variance was performed on the dependent variable of the Total Emotion Recognition performance. The analysis showed only a statistically significant Diagnostic group x Educational level interaction effect on Total Emotion Recognition, F (4, 28.910) = 4.117, *p* = 0.004, η^2^ = 0.166 (Figure 2).

Post hoc Bonferroni comparisons showed that HC significantly differed in total emotion recognition from MCI, I-J = 4.434 *p* < 0.001, and VRF, I-J = 4.391, *p* < 0.001, diagnostic groups. MCI and VRF did not differ between each other. Post Hoc comparisons regarding educational level, showed that low educational level differed significantly in total emotion recognition performance from the other two educational levels: I-J = −2.367, *p* = 0.002 and I-J = −3.179, *p* < 0.001, -for medium and high educational level, respectively. High and medium educational levels did not differ from each other.

As shown in Figure 2, the performance of MCI people is essentially low and almost the same between the three educational levels. Only the high educational level seems able to ensure a good-level-performance in VRF, while low- and medium-level-education in people with VRF does not help to avoid ER decline, which is comparable to that of MCI. Both high and medium educational levels can ensure a good-level-performance in HC. 

Subsequently, Pearson correlations were computed to examine the relationships between the three variables: recognition of Happiness, recognition of Neutral Condition, and Total Emotion Recognition. A significant correlation emerged between the Neutral Condition and Total Emotion Recognition, r = 0.498, *p* < 0.001. 

The mean scores and standard deviations in the performance of the three groups in the three variables examined in this study are given in the table below (Table 2). 

## 6. Discussion

The purpose of the study was to examine the ability to recognize emotions using dynamic visual displays in people of a specific health continuum: healthy adults of advancing age, adults with vascular risk factors and people diagnosed with Mild Cognitive Impairment. We expected that MCI patients would display the lower performance in ER—due to possible brain pathology of AD type or to the combination of this pathology and vascular pathologies (Hypothesis A).

Based on the findings, this hypothesis was partially confirmed as regards total emotion recognition. As it was shown, people with MCI performed worse compared to healthy adults of advancing age. However, their performance was not differentiated from that of the adults with vascular risk factors. At this point, it should be mentioned that MCI and VRF groups of the study were matched in age, educational level, and gender representation. Moreover, both groups had participants bearing vascular risk factors; so, the only difference seems to lie in the diagnosis of MCI carried by the first group. MCI diagnosis could explain the low emotion recognition performance of these people on the basis of hippocampal atrophy [63,64], β-amyloid deposits [65] and tau neurofibrillary tangles [66], common pathological signs that accompany behavioral and neuropsychological symptoms of MCI. It has already been shown that the areas affected in MCI are almost identical to those that mediate the recognition of emotions, especially the negative ones [64]. For the emotion of positive surprise, the recognition of which is also included in the Total Emotion Recognition score, it has also been found that regions involved in uncertain event recognition and processing, amygdala and insula as well as the parahippocampal, fusiform, and postcentral gyri [67], are directly affected in MCI pathology. Given that most people in the MCI group were aMCI, it must be noted that according to recent research aMCI subtypes with medial temporal lobe dysfunction perform worse in emotion recognition compared to other MCI subtypes; moreover, it has even been argued that emotion recognition is a sensitive and reliable indicator of their differential diagnosis [42]. 

However, beyond MCI brain pathology, given that MCI and VRF groups in the present study displayed almost the same performance in total emotion recognition, it seems that MCI people are also very likely to carry cerebral small vessel disease (SVD), which often co-exists in the pathogenesis of neurodegeneration, including Alzheimer’s disease, and in bearing vascular risk factors [68]. Vascular risk factors as causes of metabolic syndrome, can accelerate the onset and progression of cerebral small vessel dysfunctions by altering the structure and function of the blood vessels, which can lead to mild bleeding, white matter damage, and brain atrophy [69,70,71]. As has been shown, metabolic diseases of the brain bring about changes in the neuronal structure and subsequent decline in cognitive functions [72,73,74,75,76,77]. It has indeed been found that vascular risk factors and small vessel disease constitute a preclinical stage of dementia [78]. Beyond SVD, inflammation has recently been proposed as a possible pathogenic mechanism mediating the effect of vascular risk factors on the small cerebral arteries [79] implicated in neurodegeneration [80]. A recent study [42] has shown that lower education was the main predictor of increasing levels of systemic inflammation. Furthermore, an association between age, hypertension and inflammation has already been shown [81]. In conclusion, systemic inflammation is linked to heightened risk of dementia, indicating that the combination of age-related vascular risk factors, oxidative stress and endothelial dysfunction cause breakdown of the blood–brain barrierio [79].

Small Vessel Disease (SVD) also appears to be related to myelin loss [82]. A possible explanation is oligodendrocyte shrinkage originating from ischemia and cerebral hypoperfusion [83]. It has also been shown that older adults with cerebral Small Vessel Disease display white matter disruption in the genu of the corpus callosum [84], which is associated with poorer global cognitive function and executive functioning [82].

Hence, Hypothesis B of the study, according to which community dwelling adults bearing VRF would perform at a lower level, as compared to HC, seems to be confirmed. In fact, vascular pathologies that could underlie the bearing of vascular risk factors may be more appropriate candidates than MCI pathology to explain the decline of the ability to recognize emotions in both people diagnosed with MCI and community dwelling adults bearing VRF, as compared to healthy adults of advancing age. The contribution of the present study, at this point, is that the finding of low ER performance in community dwelling VRF adults can lead the experts to examine the possibility to consider emotion recognition tests as useful neuropsychological tools for detecting objective decline related to brain pathology in adults of advancing age, as soon as possible.

Regarding ER performance differences as a result of educational level, the interaction effect of education and diagnosis on the recognition of emotions has been found in some studies [5]. Sayaka Wada et al. (2021) [85] found that the years of education are associated with volumes of the right orbital gyrus, the right inferior frontal gyrus, and the right supramarginal gyrus; the right supramarginal gyrus volume is also associated with the emotion recognition ability. Beyond the neuronal level, it is well known that education contributes to crystallized intelligence, which can generally act as a compensative, protective factor against cognitive decline [86,87]; moreover, in the Greek cultural context, education can lead to higher socio-economic status and consequently, to richer life experiences. In this context, adults of advancing age with greater cognitive reserve may perform higher in emotion recognition tests as well, than their lower educated counterparts. Demenescu et al. [87], have shown that individuals with lower education display stronger amygdala responses to emotional stimuli illustrating that high education is associated with reduced amygdala responses to facial expressions and stronger amygdala—insula coupling during face perception (that is, to a brain mechanism of emotion regulation). 

Based on the findings of the present study, education was found to have this compensatory capacity only for VRF and HC groups but not for MCI. This could be due to the greater brain damage that MCI people suffer due to AD pathology. It is well known that the compensatory role of crystallized intelligence is limited when fluid intelligence that depends on the biology of the brain gets seriously impaired [88]. In the same vein, the finding that more of 13 years of education (high educational level) are needed to compensate for brain damage related to VRF, while even 10 to 12 years of education can be enough for HC, is also indicative of the following; the greater the brain damage, the more years of education it takes to compensate for emotion recognition decline. 

As regards the almost excellent performance of all groups in happiness recognition, Bora et al. (2017) [89] support that happiness recognition is indeed preserved even in adults diagnosed with MCI. Pietschnig et al. (2016) [7] report preservation of this ability in aMCI as well. The same findings for the recognition of Happiness in patients with Alzheimer’s dementia were reached by Guaita 2009 [90] and Soowon Park (2017) [2] who also found perfect ability to recognize happiness in patients with Frontotemporal dementia.

The findings of these studies indicate that recognition of happiness may not share the same neural network with the recognition of other emotions [91]; it has also been argued that the neurons that mediate the perception of happiness are not those affected by age and vascular burden [92,93]. Another perspective holds that older adults tend to focus on more positive aspects of a situation due to their limited time perspective, compared to younger adults who tend to display a biologically defined “negativity bias” (It seems that older adults, perceiving the end of their lives and valuing their remaining time, prefer emotional goals that could enhance and optimize their current experience. In this way, they tend to evaluate a situation more positively, especially when not enough information is provided either to the contrary or to the absence of an emotion [94,95,96,97]. It has also been argued that the ability of the positive investment of older adults requires at least a minimum level of cognitive control abilities [98]. There are main arguments to further support this view; older adults with well-maintained cognitive abilities are more effective in emotion regulation compared to adults with impaired cognitive abilities. Moreover, when the process of recognizing an emotional state interferes with the goal—i.e., splitting attention between multiple tasks—older adults tend to have a negative perception of the situation [99,100]. Cognitive control is mediated by the prefrontal cortex, brain area sensitive to the effects of neurodegenerative disorders [97,101]. In this vein and according to the findings of the present study, MCI patients and VRF adults seem to maintain the necessary cognitive reserve in terms of cognitive control, able to support recognition of happiness [102].

Regarding the recognition of the emotionally neutral condition, a trend was observed for diagnosis and age-group to negatively affect performance. According to Mc Donald et al. [60], there is a more general difficulty in recognizing the neutral condition of EET because of its strongly ambiguous nature. This finding was also supported by the findings of Moraitou et al. (2013) for the Greek adaptation [61]. Hence, the ambiguity of the scenes (they were usually considered anxiety conditions [61]) related to neutral condition may be a methodological issue in need of further examination. However, the positive correlation between recognition of neutral condition and total emotion recognition may be due to difficulties in processing uncertain situations in general, something that requires a good level of cognitive control [103].

### 6.1. Conclusions

Summarizing, the present study showed that emotion recognition ability declines in MCI and can also decline at the same rate in community dwelling adults bearing vascular risk factors, as compared to healthy adults of advancing age. Thus, it could be useful for the clinicians to administer emotion recognition tests in the broader neuropsychological assessment context, to find as soon as possible objective indices of brain-related decline. A second interesting finding is that, to a certain extent, education can compensate for emotion recognition decline which is defined by the magnitude of brain damage.

### 6.2. Limitations and Future Research

The study has limitations. The first author, who is the only examiner, was not blinded to the clinical diagnosis of the three groups. The sample was convenient and healthy controls were younger and more educated than the others. However, it is extremely difficult to find healthy older adults in Greece. In addition, the two genders are not equally represented in the study, as there are many more women, which could affect the results of the present study. In terms to the pathological groups; MCI and adults with VRF, as clinical entities, show heterogeneity in their characteristics, a fact that in future research would be useful to take into account, aiming to a stricter selection of participants so that there are more explicit results.

The study is cross-sectional. Future research can have a longitudinal design to study emotion recognition decline in the health continuum ‘healthy aging—vascular aging—MCI’ in the same participants by examining them many times during their lifespan development and aging. Specific subtypes of MCI or vascular aging can be assessed in emotion recognition to study similarities and differences. Other dynamic tests designed to measure emotion recognition can also be administered in a new study, to confirm (or not) the results.

## Figures and Tables

**Figure 1 ijerph-19-13366-f001:**
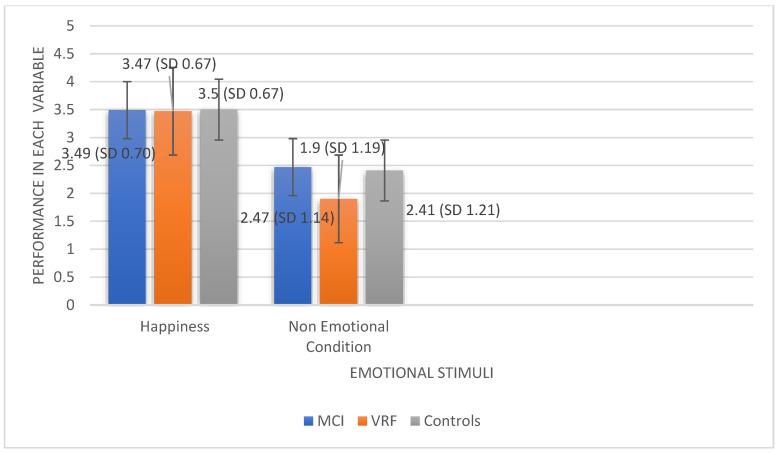
Mean scores of the three diagnostic groups in Recognition of Happiness and Emotionally Neutral Condition.

**Figure 2 ijerph-19-13366-f002:**
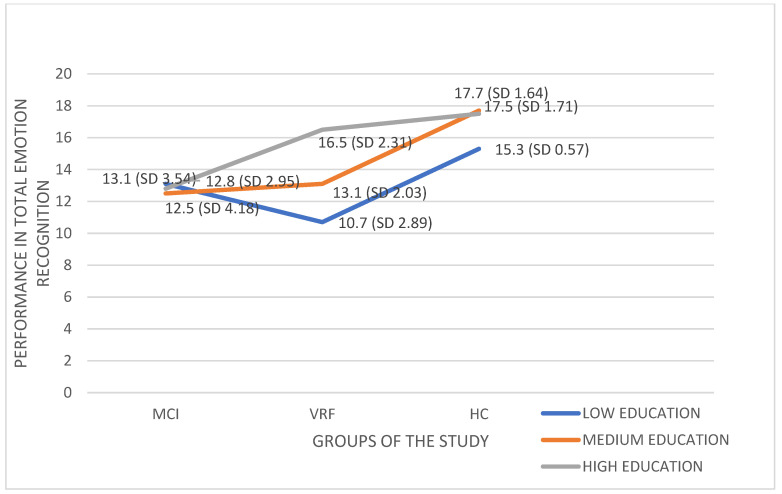
The effects of Diagnostic group x Educational level interaction on Total Emotion Recognition performance.

**Table 1 ijerph-19-13366-t001:** Age, Education and Gender representation for the three diagnostic groups.

Groups	MCI (n = 43)	VRF (n = 41)	HC (n = 22)
Age-groups: Late Middle-Aged Adults/Older Adults	12/31	14/27	22/0
Educational Level: High/Middle/Low	17/11/15	12/8/21	10/9/3
Gender: Men/Women	9/34	9/32	4/18

**Table 2 ijerph-19-13366-t002:** Mean Scores and Standard Deviations of the study variables (Happiness, Non Emotional Condition, Total Emotion Recognition) for the study population (MCI, adults with VRF and Healthy Controls).

	Happiness	Non Emotional Condition	Total Emotion Recognition
MCI	3.49 (S.D. 0.703)	2.47 (S.D. 1.14)	12.88 (S.D. 2.95)
Adults with VRF	3.47 (S.D. 0.679)	1.90 (S.D. 1.19)	12.92 (S.D. 2.03)
Healthy Controls	3.50 (S.D. 0.673)	2.41 (S.D. 1.21)	13.82 (S.D. 3.66)

## Data Availability

Not Applicable.

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
