# Peer review of "Emotion Recognition in a Health Continuum: Comparison of Healthy Adults of Advancing Age, Community Dwelling Adults Bearing Vascular Risk Factors and People Diagnosed with Mild Cognitive Impairment"

_ijerph, 2022, doi:10.3390/ijerph192013366_

Round 1

Reviewer 1 Report

Comments to the Authors

In this study, Tsentidou and colleagues investigated emotion recognition ability in adults with MCI, adults bearing one or more VRF, and healthy controls of advancing age (HC). They administered a dynamic visual test to examine recognition 23 of basic emotions and emotionally neutral conditions. Significant differences were observed in diagnostic group x educational level interaction as regards total emotion recognition ability. They showed that the high educational level seems to contribute to a high-level emotion-recognition performance both in healthy adults of advancing age and in adults bearing vascular risk factors whereas, the medium educational level appears to play the same role only in healthy adults.  Furthermore, they showed that neither educational level can help MCI people to enhance their significantly lower emotion recognition ability. This study is of some interest and might contribute to a better understanding effect on emotion recognition ability of vascular risk factors. However, I have comments and suggestions given below.

Introduction

P. 2, L. 68-81: In this paragraph, it presented impaired emotion recognition in neurodegenerative diseases (Alzheimer’s patients, etc.) by behavioral and MRI studies. I would also recommend adding electrophysiological studies in which the patients with neurodegenerative diseases showed impaired and different emotion recognition abilities compared to healthy subjects (as well as behavioral and MRI studies). Briefly, it can be mentioned electrophysiological studies (for example EEG studies).

Materials and Methods

P. 3, L. 131-135: In this sentence, it was given exclusion criteria of the study. Did the authors consider the use of psychological drugs, history of substance abuse or alcoholism, and/or history of psychiatric disease, etc. for the exclusion criteria? These should be added to the exclusion criteria.

P. 4, L. 190-192: In the study, no difference arose in the gender of the groups. However, the number of females was much in all groups. It is known that gender effect on emotion. There are differences between females and males in emotion recognition. Therefore, this should be added as a limitation of the study.

Results

For better and easier understanding, I suggest that is given a table that showed the results of the study in the result section.

The figures (Figure 1 and 2) are very simple in the results section, it can be improved more.

Discussion

No comment

Author Response

Dear Reviewer, thank you very much for the time you have devoted to our manuscript and for your valuable suggestions that improve it.

Comment 1: «P. 2, L. 68-81: In this paragraph, it presented impaired emotion recognition in neurodegenerative diseases (Alzheimer’s patients, etc.) by behavioral and MRI studies. I would also recommend adding electrophysiological studies in which the patients with neurodegenerative diseases showed impaired and different emotion recognition abilities compared to healthy subjects (as well as behavioral and MRI studies). Briefly, it can be mentioned electrophysiological studies (for example EEG studies).».

Reply: Thank you very, much for this note. Relevant studies have been added to the introduction. Please see lines 68-72.

Comment 2: «P. 3, L. 131-135: In this sentence, it was given exclusion criteria of the study. Did the authors consider the use of psychological drugs, history of substance abuse or alcoholism, and/or history of psychiatric disease, etc. for the exclusion criteria? These should be added to the exclusion criteria.».

Reply: Thank you very much for this reminder. Of course, mental and/or psychiatric diseases, alcoholism and/or drug use disorders were exclusion criterions for the study population. Please see lines 141-143.

Comment 3: «P. 4, L. 190-192: In the study, no difference arose in the gender of the groups. However, the number of females was much in all groups. It is known that gender effect on emotion. There are differences between females and males in emotion recognition. Therefore, this should be added as a limitation of the study.».

Reply: Thank you very much for the underlining. Indeed, women are overrepresented in the study, however this is also the case in the population of adults diagnosed with Mild Cognitive Impairment. It has been added to the limitations of the study. Please see lines 443-445.

Comment 4: «For better and easier understanding, I suggest that is given a table that showed the results of the study in the result section.».

Reply: Thank you very much for this note. Please see lines 310 -315.

Comment 5: «The figures (Figure 1 and 2) are very simple in the results section, it can be improved more.».

Reply: Thank you very much for this comment. Please see Figure 1 and 2.

Reviewer 2 Report

RE: ijerph-1925243

The authors have assessed emotional recognition (ER) in three convenience samples of older persons recruited from Memory Clinics in Thessaloniki, Greece, outpatient clinics of the General Hospital of Katerini, Greece, and healthy adult volunteers from the wider community of Pieria, Greece. ER was assessed by the Emotion Evaluation Task (EET) of The Awareness of Social Inference Test (TASIT). Cross group differences were tested by non-parametric MANOVA w/ Bonferroni correction.

The assessment of ER is a potentially interesting issue related to the social integration of older persons.

 The main problems with this analysis are the small sample size, baseline cross-group differences in age and education, and selection of the groups for comparison. The small sample size and cross group differences force their use of non-parametric statistical approaches, weakening power and increasing the risk of negative findings. A power analysis should have been performed to show power to detect an effect, and therefore validate the negative findings.

 Another issue is the interpretation of these convenience samples as though they are presenting across some continuum of “health”. “MCI” is a heterogenous construct. One can be labeled as a member of that group with impairments from any of a wide range of non-dementing cognitive impairments, only some of which might relate to ER. Similarly, subclinical vascular disease might impact a number of cognitive domains depending on the distribution and location of the ischemic lesions, and variable impact ER. The “healthy volunteers” could have been restricted to an older age frams (e.g., spouses of the other two group participants, to reduce the risk of cross-group differences that can impact the analysis of such a small sample.

Author Response

Dear Reviewer, thank you very much for the time you have devoted to our manuscript and for your valuable suggestions that improve it.

Comment 1: «The main problems with this analysis are the small sample size, baseline cross-group differences in age and education, and selection of the groups for comparison. The small sample size and cross group differences force their use of non-parametric statistical approaches, weakening power and increasing the risk of negative findings. A power analysis should have been performed to show power to detect an effect, and therefore validate the negative findings.».

Reply: «Thank you very much for this note. Initially the authors argue that parametric statistical tests are the best type of measures, and they were used in the present study; ANOVA and MANOVA. Power analysis showed that the required simple size = 102. Thus, our sample size seems to be enough.

Comment 2: « Another issue is the interpretation of these convenience samples as though they are presenting across some continuum of “health”. “MCI” is a heterogenous construct. One can be labeled as a member of that group with impairments from any of a wide range of non-dementing cognitive impairments, only some of which might relate to ER. Similarly, subclinical vascular disease might impact a number of cognitive domains depending on the distribution and location of the ischemic lesions, and variable impact ER. The “healthy volunteers” could have been restricted to an older age frams (e.g., spouses of the other two group participants, to reduce the risk of cross-group differences that can impact the analysis of such a small sample.».

Reply: «Thank you very much for this underline. The authors of the present study reviewed very recent literature suggesting that adults with vascular risk factors are actually a stage before the onset of cognitive deficits and progression to MCI (Please see indicatively, Arfanakis, K; Evia, A.M; Leurgans, S.E; Cardoso, L; Kulkarni, A; Alqam, N; Lopes, L.F; Vieira, D; Bennett, D.A; Schneider, J.A. Neuropathologic Correlates of White Matter Hyperintensities in a Community-Based Cohort of Older Adults. J Alzheimer's dis: JAD, 2020,73(1), 333–345. https://doi.org/10.3233/JAD-190687., Wardlaw, J.M; Valdés Hernández, M.C; Muñoz-Maniega, S; What are white matter hyperintensities made of? Relevance to vascular cognitive impairment. J. Am. Heart Assoc. 2015, 4(6), 001140. https://doi.org/10.1161/JAHA.114.001140., Olsen, R.K; Yeung, L.K; Noly-Gandon, A; D’Angelo, M.C; Kacollja, A; Smith, V.M; Ryan, J.D; Barense, M.D. Human anterol-ateral entorhinal cortex volumes are associated with cognitive decline in aging prior to clinical diagnosis. Neurobiol Aging 2017,57, 195–205. doi: 10.1016/J.NEUROBIOLAGING.2017.04.025., Zhang, Y; Zhang, X; Zhang, J; Liu, C; Yuan, O; Yin, X; Wei, L; Cui, J; Tao, R; Wei, P; Wang, J. Gray matter volume abnormalities in type 2 diabetes mellitus with and without mild cognitive impairment. Neurosci. Lett.  2014, 562, 1–6., Groeneveld, O; van den Berg, E; Rutten, G; Koekkoek, P; Kappelle, J; Biessels, G. Applicability of diagnostic constructs for cognitive impairment in patients with type 2 diabetes mellitus, Diab Res Cli  Pract 2018, 142, 91-99., Gocmeza, S; Åžahina, T; Yazirb, Y; Duruksuc, G; Eraldemird, F; Polatc, S; Utkan, T. Resveratrol prevents cognitive deficits by attenuating oxidative damage and inflammation in rat model of streptozotocin diabetes induced vascular dementia Physiology & Behav 2019, 201, 198–207., Rosenberga, J; Lecheaa, N; Pentanga, G; Shaha, N. What magnetic resonance imaging reveals – A systematic review of the relationship between type II diabetes and associated brain distortions of structure and cognitive functioning Front Neuroen-docrinol. 2019, 52, 79–112., Kivipelto, M; Helkala, L; Laakso, P; Hanninen, T; Hallikainen, M; Alhainen, K; Soininen, H. Apolipoprotein E ε4 allele, el-evated midlife total cholesterol level, and high midlife systolic blood pressure are independent risk factors for late-life Alzheimer disease. Ann Inter Med. 2002, 137(3), 149–155.)

We would also like to mention that the patients with MCI who took part in the study were of amnestic type; hence, their main problem was in memory function but of course they suffer from impairment in other domains (mostly, executive functions); as for adults with cardiovascular risk factors, are indeed a heterogeneous group. However, the aim of cognitive neuropsychology is exactly to find a diagnostic measure that can “capture” cognitive impairment as soon as possible in as many individuals as possible of these two pathologies besides their heterogeneity. We also mention this issue into the limitations.  Please see lines 452-455.

Regarding the healthy group, a great deal of effort was made, and a considerable amount of study time was consumed in identifying a group equal to the pathological groups, but unfortunately for the data of Greece (and of other countries of southern Europe such as Italy, please see the citations), this is not possible. As regards the relatives of the patients, they have been evaluated and they certainly cannot be considered healthy, since the vast majority of them show emotional difficulties (anxiety or even mood disorders) and have accompanying vascular risk factors.

Round 2

Reviewer 1 Report

The authors have addressed my remarks adequately, and I have no other comments.

Author Response

Dear Reviewer, thank you very much for the time you have devoted to our manuscript and for your valuable suggestions that improve it.